# Learning Graph Convolution Filters from Data Manifold

## Abstract

Convolution Neural Network (CNN) has gained tremendous success in computer vision tasks with its outstanding ability to capture the local latent features. Recently, there has been an increasing interest in extending CNNs to the general spatial domain. Although various types of graph convolution and geometric convolution methods have been proposed, their connections to traditional 2D-convolution are not well-understood. In this paper, we show that depthwise separable convolution is a path to unify the two kinds of convolution methods in one mathematical view, based on which we derive a novel Depthwise Separable Graph Convolution that subsumes existing graph convolution methods as special cases of our formulation. Experiments show that the proposed approach consistently outperforms other graph convolution and geometric convolution baselines on benchmark datasets in multiple domains.

## 1 Introduction

Convolution Neural Network (CNN) (LeCun et al., 1995) has been proven to be an efficient model family in extracting hierarchical local patterns from grid-structured data, which has significantly advanced the state-of-the-art performance of a wide range of machine learning tasks, including image classification, object detection and audio recognition (LeCun et al., 2015). Recently, growing attention has been paid to dealing with data with an underlying graph/non-Euclidean structure, such as prediction tasks in sensor networks (Xingjian et al., 2015), transportation systems (Li et al., 2017), and 3D shape correspondence application in the computation graphics (Bronstein et al., 2017). How to replicate the success of CNNs for manifold-structured data remains an open challenge.

Many graph convolution and geometric convolution methods have been proposed recently. The spectral convolution methods (Bruna et al., 2013; Defferrard et al., 2016; Kipf & Welling, 2016) are the mainstream algorithm developed as the graph convolution methods. Because their theory is based on the graph Fourier analysis (Shuman et al., 2013), one of their major limitations is that in this model the knowledge learned from different graphs is not transferrable (Monti et al., 2016). Other group of approaches is geometric convolution methods, which focuses on various ways to leverage spatial information about nodes(Masci et al., 2015; Boscaini et al., 2016; Monti et al., 2016). Existing models mentioned above are either not capable of capturing spatial-wise local information as in the standard convolution, or tend to have very large parameter space and hence, are prone to overfitting. As a result, both the spectral and the geometric convolution methods have not produced the results comparable to CNNs on related tasks. Such a misalignment makes it harder to leverage the rapidly developing 2D-convolution techniques in the generic spatial domain. We note graph convolution methods are also widely used in the pure graph structure data, like citation networks and social networks (Kipf & Welling, 2016). Our paper will only focus on the data with the spatial information.

In this paper, we provide a unified view of the graph convolution and traditional 2D-convolution methods with the label propagation process (Zhu et al., 2003). It helps us better understand and compare the difference between them. Based on it, we propose a novel Depthwise Separable Graph Convolution (DSGC), which inherits the strength of depthwise separable convolution that has been extensively used in different state-of-the-art image classification frameworks including Inception Network (Szegedy et al., 2016), Xception Network (Chollet, 2016) and MobileNet (Howard et al., 2017). Compared with previous graph and geometric methods, the DSGC is more expressive and

aligns closer to the depthwise separable convolution network, and shares the desirable characteristic of small parameter size as in the depthwise separable convolution. In experiments section, we evaluate the DSGC and baselines in three different machine learning tasks. The experiment results show that the performance of the proposed method is close to the standard convolution network in the image classification task on CIFAR dataset. And it outperforms previous graph convolution and geometric convolution methods in all tasks. Furthermore, we demonstrate that the proposed method can easily leverage the advanced technique developed for the standard convolution network to enhance the model performance, such as the Inception module (Szegedy et al., 2016), the DenseNet architecture (Huang et al., 2016) and the Squeeze-and-Excitation block (Hu et al., 2017).

The main contribution of this paper is threefold:

- A unified view of traditional 2D-convolution and graph convolution methods by introducing depthwise separable convolution.
- A novel Depthwise Separable Graph Convolution (DSGC) for spatial domain data.
- We demonstrate the efficiency of the DSGC with extensive experiments and show that it can facilitate the advanced technique of the standard convolution network to improve the model performance.

## 2 A GRAPH PERSPECTIVE OF CONVOLUTION

We provide a unified view of label propagation and graph convolution by showing that they are different ways to aggregate local information over the graphs or data manifolds. We then discuss connections between graph convolution and depthwise separable convolution over the 2D-grid graph, which motivates us to propose a new formulation that subsumes both methods as special cases.

Unless otherwise specified, we denote a matrix by $X$, the $i$-th row in the matrix by $x_i$, and $(i, j)$-th element in the matrix by $x_{ij}$. Superscripts are used to distinguish different matrices when necessary. All the operations being discussed below can be viewed as a function that transforms input feature maps $X \in \mathbb{R}^{N \times P}$ to output feature maps $Y \in \mathbb{R}^{N \times Q}$, where $N$ is the number of nodes in the graph and $P, Q$ are the number of input and features (channels) associated with each node respectively. We use $\mathcal{N}(i)$ to denote the set of neighbors for $i$-th node.

### 2.1 LABEL PROPAGATION

Label propagation (LP) (Zhu et al., 2003) is a classic approach to aggregate local information over a graph. The basic version of LP can be written as

$$y_{iq} = \sum_{j \in \mathcal{N}(i)} w_{ij} x_{jq} \tag{1}$$

where $W$ is a normalized adjacency matrix that summarizes the graph structure. The intuition is that the value of node $i$ is updated via a weighted combination of its neighbors.

### 2.2 GRAPH CONVOLUTION

Graph convolution (Kipf & Welling, 2016) (GC) is a recently proposed graph convolution operator that can be viewed as an extension of LP, formulated as

$$y_{iq} = \sum_{j \in \mathcal{N}(i)} w_{ij} z_{jq} \quad where \quad z_j = U x_j \tag{2}$$

where $W$ is a symmetrically normalized adjacency matrix with a ridge on its diagonal, which is a deterministic matrix given the input data, and $U \in \mathbb{R}^{P \times Q}$ represents a linear transformation. Following the Chollet (2016), $W$ is named as the spatial filter and $U$ is named as the channel filter. The original form of graph convolution, such as the Spectral Network (Bruna et al., 2013), is derived from graph signal processing (Shuman et al., 2013) as a generalization of Fourier analysis to the domain of graphs. Several limitations of the Spectral Network, such as its high computation

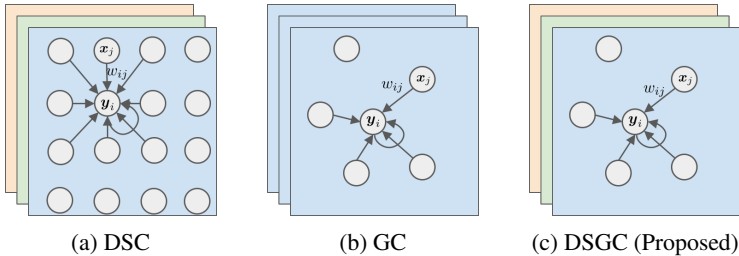

| (a) DSC | (b) GC | (c) DSGC (Proposed) |

Figure 1: Visualization of different convolution operations with three output channels. We use different colors to represent different filters (weight configurations of the links). (a) DSC defined on 2D grid graphs. (b) GC defined on generic graphs. (c) DSGC defined on generic graphs.

complexity and the lack of locality, are addressed in Defferrard et al. (2016) (ChebyNet) and further refined by Kipf & Welling (2016) via approximation.

To Compare LP with GC, the former only utilizes the graphical information, while the latter has an additional linear transformation of $x_j$ to into the intermediate representation $z_j$ via matrix $U$. This additional step makes GC capable of capturing the dependencies among features (channels), which yields performance improvement.

### 2.3 DEPTHWISE SEPARABLE CONVOLUTION

For a full 2d-convolution layer, the convolution filters encode channel correlation and spatial correlation simultaneously (Chollet, 2016). Then depthwise separable convolution (DSC) is proposed under the intuition that the channel correlation and spatial correlation could be decoupled, and has been found successful in several modern architectures for image classification (Chollet, 2016). We choose to focus on DSC (instead of full convolution) because of its strong empirical performance with a small number of parameters, and its intimate connections to GC which will be revealed in the following. And we discuss the full convolution formulation with the label propagation process in Section 5.

By viewing each pixel in the image as a node, DSC can be formulated in a graph-based fashion

$$y_{iq} = \sum_{j \in \mathcal{N}(i)} w^{(q)}_{\Delta_{ij}} z_{jq} \quad where \quad \boldsymbol{z}_j = \boldsymbol{U}\boldsymbol{x}_j \tag{3}$$

where $\Delta_{ij}$ denotes the relative position of pixel $i$ and pixel $j$ on the image, and $w^{(q)}$ can be viewed as a lookup table with the pixel-pixel offset $\Delta_{ij}$ as the key, according to the stationarity (weight-sharing) assumption of convolution. In the context of images, $\mathcal{N}(i)$ denotes the index set of surrounding pixels for $i$-th pixel, which is equivalent to the $k$-nearest neighbor set under the Euclidean distant metric. For example, the size of $\mathcal{N}(i)$, or $k$, is 9 for a $3 \times 3$ convolution filter (considering self-loop).

## 3 PROPOSED METHOD

### 3.1 DEPTHWISE SEPARABLE GRAPH CONVOLUTION

We notice that the formulation of GC and DSC is similar except that

1. Spatial filters in DSC are channel-specific, while GC uses a global spatial filter.
2. Spatial filters in DSC are learned from the data (under the stationarity constraints), while the filter in GC is a constant matrix with the given input.

On the one hand, DSC does not apply to the domain of generic spatial data lying on the manifold where the space of $\Delta_{ij}$ (defined as the difference of the spatial coordinates between node $i$ and

node $j$) can be infinite. On the other hand, GC suffers from the restriction that all channels have to share the same given spatial filter. This heavily constrains the model capacity, which would be more severe when the deeper network structure is used. In the context of graphs, it would be desirable to have multiple spatial filters—to capture a diverse set of diffusion patterns over the graph or data manifold, which is the same as the convolution filters in the image domain.

To address these limitations, we propose Depthwise Separable Graph Convolution (DSGC) which naturally generalizes both GC and DSC

$$y_{iq} = \sum_{j \in \mathcal{N}(i)} w^{(q)}(\Delta_{ij}) z_{jq} \quad where \quad \boldsymbol{z}_j = \boldsymbol{U} \boldsymbol{x}_j \tag{4}$$

where we slightly abuse the notation by overloading $w^{(q)}(\cdot)$ as a function, which maps $\Delta_{ij}$ to a real number, and $\mathcal{N}(i)$ still represents the $k$-nearest neighbor sets. To understand the proposed formulation, notice

1. Different from DSC, the stationarity requirement is implemented in a "soft" manner by defining a function instead of by the set of equality constraints. In our experiment, each $w^{(q)}(\cdot)$ is a function parameterized by a two-layer MLP.

2. Different from GC, channel-specific convolution is enabled by learning multiple spatial convolution filters. This amounts to simultaneously constructing multiple graphs under the different node-node similarity metrics, where the metrices are implicitly defined by neural networks and hence, are jointly optimized during the training.

Overfitting is a common issue in graph-based applications, due to limited data available. To alleviate this issue, we propose an option to group the channels into $C$ groups, where $D = Q/C$ channels in the same group would share the same filter.

$$w^{(q)}(\cdot) = w^{(q')}(\cdot) \quad if \quad \lfloor \frac{q}{D} \rfloor = \lfloor \frac{q'}{D} \rfloor \tag{5}$$

## 3.2 Normalization

The context of each node in any given generic graph, namely its connection pattern with neighbors, can be non-stationary over different parts of the graph, while it is constant in the 2d-grid graphs. It is, therefore, a common practice to normalize the adjacency matrix in order to make the nodes adaptive to their own contexts (Eq.1). A natural way to carry out normalization for DSGC is to apply a softmax function over the predicted spatial filter weights at each node, which can be written as $\tilde{\boldsymbol{w}}_i = softmax(\boldsymbol{w}_i)$, where $\boldsymbol{w}_i$ stands for the $i$-th row of spatial filter $\boldsymbol{W}$ learned by a neural network. We empirically find normalization leads to better performance and significantly speeds up the convergence.

In the following experiments, we use the proposed depthwise separable graph convolution with a linear highway bypass as the basic convolution component and imitate the rest setting of the standard convolution neural network to solve different machine learning tasks.

## 4 Experiments

### 4.1 Experiment Setting

We evaluate the proposed Depthwise Separable Graph Convolution (DSGC) method with representative baselines in the prediction tasks of image classification, time series forecasting, and document categorization. The algorithms are implemented in PyTorch; all the data and the code are made publicly accessible [1]. For controlled experiments, all the graph convolution methods share the same empirical settings unless otherwise specified, including network structures, the dimension of latent factors, and so on. The optimization algorithm is applied to all models. The neural network used to model the spatial convolution filter ($w^{(q)}(\cdot)$) in Eq.4 is a two-layers MLP with 256 hidden dimension and tanh activation function. We have conducted ablation tests with the two-layer MLP by

---

[1]Code: Data: the links are anonymous due to the double-blind policy.

changing the number of layers and activation function of each hidden layer, and by trying several weight sharing strategies. The results are very similar; the two-layer MLP provides a reasonable performance with the shortest running time. Appendix A contains more details, such as the network architecture and model hyper-parameters.

## 4.2 EVALUATION ON IMAGE CLASSIFICATION

We conduct experiments on CIFAR10 and CIFAR100 (Krizhevsky & Hinton, 2009), which are popular benchmark datasets in image classification. Both sets contain 60000 images with $32 \times 32$ pixels but CIFAR10 has 10 category labels and CIFAR100 has 100 category labels. Each image is typically treated as a $32 \times 32$ grid structure for standard image-based convolution. To enable the comparison on generic graphs, we create the modified versions of CIFAR10 and CIFAR100, respectively, by subsampling only 25% of the pixels from each graph. As illustrated in Figure 2, the subsampling results in irregularly scattered nodes for each image.

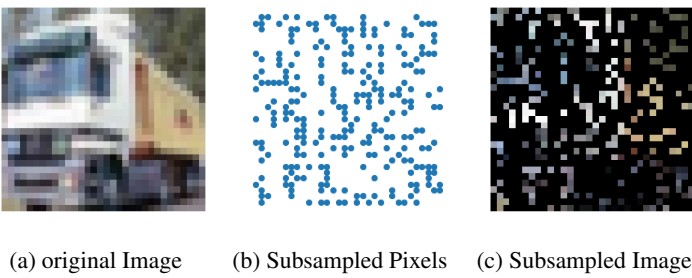

(a) original Image     (b) Subsampled Pixels     (c) Subsampled Image

Figure 2: How to construct subsampled CIFAR datasets: (a) is an example image from CIFAR dataset. (b) is the subsampled pixels map. The blue points indicate which points are sampled. (c) is the image after sampling, where the black points are those being sampled out.

For comparison we include the traditional 2d convolution and graph convolution networks as baselines, including standard CNN; Xception network (Chollet, 2016) which uses the depthwise separable convolution; DCNN (Atwood & Towsley, 2016), the method using multi-hops random walk as the graph filters; ChebyNet (Defferrard et al., 2016), the method using Chebyshev polynomial to approximate the Fourier transformation of (irregular) graphs; GCN (Kipf & Welling, 2016) which is described in Section 2; MoNet (Monti et al., 2016), the method using Gaussian function to define the propagation weights over (irregular) graphs. For a fair comparison, we use the VGG13 architecture (Simonyan & Zisserman, 2014) in all the methods above as the basic platform, and replace the convolution layers according to the methods. The pooling layer is performed by the kmean clustering. The centroid of each clusters is regarded as the new node after pooling, and its hidden vector is the mean or max over the nodes in the cluster, based on the pooling method. Notice that, we only normalize the input signals to [0,1] and do not have other preprocessing or data augmentation.

The experiment results are summarized in Table 1. Firstly, we observe that Xception and CNN have the best results; this is not surprising because both methods use grid-based convolution which is naturally suitable for image recognition. Secondly, DSGC outperforms all the other graph-based convolution methods, and its performance is very close to that of the grid-based convolution methods. Furthermore, contributed by the depthwise separable convolution and sharing graph technique, our model can achieve the competitive performance without increasing the number of parameters as GCN, the one with the smallest number of parameters among the graph convolution approaches. In appendix A.4, we further report the variance of DSGC model, which shows the improvement is significant and stable.

## 4.3 EVALUATION ON TIME SERIES FORECASTING

As another important application domain, here we are interested in how to effectively utilize the locality information about sensor networks in time series forecasting. For example, how to incorporate the longitudes/latitudes of sensors w.r.t. temporal cloud movement is an important question in spatiotemporal modeling for predicting the output of solar energy farms in the United States. Appendix A provides the formal definition of this task.

| | Subsampled Graphs | | | Original Graphs | | |
|---|---|---|---|---|---|---|
| Dataset | CIFAR10 | CIFAR100 | P | CIFAR10 | CIFAR100 | P |
| DCNN (Atwood & Towsley, 2016) | 43.68% | 76.65% | 12M | 55.56% | 84.16% | 50M |
| ChebyNet (Defferrard et al., 2016) | 25.04% | 49.44% | 10M | 12.99% | 36.96% | 19M |
| GCN (Kipf & Welling, 2016) | 26.78% | 51.30% | 5.6M | 19.09% | 41.64 % | 9.8M |
| MoNet (Monti et al., 2016) | 21.20% | 47.87% | 11M | 8.34% | 29.56% | 20M |
| DSGC | 18.72% | 44.33% | 5.7M | 7.31% | 27.29% | 9.9M |
| CNN (Simonyan & Zisserman, 2014) | 18.03% | 43.42% | 18M | **6.86**% | 26.86% | 18M |
| Xception (Chollet, 2016) | **17.07**% | **41.54**% | 3.1M | 7.08% | **26.84**% | 3.1M |

Table 1: Test-set error rates: P is the number of parameters

We choose three publicly available benchmark datasets for this task:

- The U.S Historical Climatology Network (USHCN)[2] dataset contains daily climatological data from 1,218 meteorology sensors over the years from 1915 to 2000. The sequence length is 32,507. It includes five subsets, and each has a climate variable: (1) maximum temperature, (2) minimum temperature, (3) precipitation, (4) snowfall and (5) snow depth. We use the daily maximum temperature data and precipitation data, and refer them as the **USHCN-TMAX** and **USHCN-PRCP** sets, respectively.

- The solar power production records in the year of 2006[3] has the data with the production rate of every 10 minutes from 1,082 solar power stations in the west of the U.S. The sequence length is 52,560. We refer this set of data as **Solar**.

All the datasets have been split into the training set (60%), the validation set (20%) and the test set (20%) in chronological order.

All the graph convolution methods (DCNN, ChebyNet, GCN and MoNet) in the previous section (Section 4.2) are included to form the baselines for comparison. We also add traditional methods for time series forecasting, such as (1) Autoregressive model (AR) which predicts future signal using a window of historical data based on a linear assumption about temporal dependencies, (2) Vector autoregressive model (VAR) which extends AR to the multivariate version, namely, the input is the signals from all sensors in the history window, and (3) the LSTNet deep neural network model (Lai et al., 2017) which combines the strengths of CNN, RNN and AR. None of those methods is capable of leveraging locational dependencies via graph convolution. We exclude the CNN and Xception methods, the 2D-grid based convolution, which could not be generalized to irregular graphs which we focus here.

Table 2 summarizes the evaluation results of all the methods, where the performance is measured using the Root Square Mean Error (RMSE). The best result on each dataset is highlighted in boldface. The first chunk of three methods does not leverage the spatial or locational information in data. The second chuck consists of the neural network models which leverage the spatial information about sensor networks. The graph convolution methods in the second chunk clearly outperforms the methods in the first chunk, which does not explicitly model the spacial correlation within sensor networks. Overall, our proposed method (DSGC) has the best performance on all the datasets, demonstrating its strength in capturing informative local propagation patterns temporally and specially.

## 4.4 DOCUMENT CATEGORIZATION

For the application to text categorization we use the 20NEWS dataset (Joachims (1996)) for our experiments. It consists of 18,845 text documents associated with 20 topic labels. Individual words in the document vocabulary are the nodes in the graph for convolution. Each node also has its word embedding vector which is learned by running the Word2Vec algorithm (Mikolov et al. (2013)) on this corpus. Following the experiment settings in Defferrard et al. (2016) we select the top 1000 most frequent words as the nodes. Table 3 summarizes the results of the graph convolution methods plus three popular traditional classifiers (Linear SVM, Multivariate Naive Bayes and Softmax). DSGC has the best result on this dataset. Notice that the traditional classifiers are trained and tested with

---

[2]http://cdiac.ornl.gov/epubs/ndp/ushcn/daily_doc.html
[3]http://www.nrel.gov/grid/solar-power-data.html

| Dataset | USHCN-TMAX | USHCN-PRCP | Solar |
|---|---|---|---|
| AR | 8.2354 | 30.3825 | 0.03195 |
| VAR | 17.9743 | 29.2597 | 0.03296 |
| LSTNet (Lai et al., 2017) | 10.1973 | 29.0624 | 0.02865 |
| DCNN (Atwood & Towsley, 2016) | 6.5188 | 29.0424 | 0.02652 |
| ChebyNet (Defferrard et al., 2016) | 5.5823 | 27.1298 | 0.02531 |
| GCN (Kipf & Welling, 2016) | 5.4671 | 27.1172 | 0.02512 |
| MoNet (Monti et al., 2016) | 5.8263 | 26.8076 | 0.02564 |
| DSGC | **5.1738** | **25.8228** | **0.02453** |

Table 2: Time series prediction: Experiment result in terms of RMSE.

the feature set of the top 1000 words, which is the same setting as in the graph convolution models. If all words are used, traditional classifiers would have higher performance.

| Method | Accuracy |
|---|---|
| Linear SVM[†] | 65.90% |
| Multinomial Naive Bayes[†] | 68.51% |
| Softmax[†] | 66.28% |
| FC2500[†] | 64.64% |
| FC2500-FC500[†] | 65.76% |
| DCNN (Atwood & Towsley, 2016) | 70.35% |
| ChebyNet (Defferrard et al., 2016) | 70.92% |
| GCN (Kipf & Welling, 2016) | 71.01% |
| MoNet (Monti et al., 2016) | 70.60% |
| DSGC | **71.88**% |

Table 3: Accuracy on the validation set. The results with [†] come from Defferrard et al. (2016).

## 4.5 DSGC Variants with Advanced Convolution Architectures

The proposed convolution method (DSGC) can be considered as an equivalent component to the depthwise separable convolution method. Naturally, we can leverage the technique developed for the standard convolution network to improve the DSGC framework. Hence we examine DSGC with the following techniques which are popular in recent years for standard convolution over images: (1) Inception module (Szegedy et al., 2016), (2) DenseNet framework (Huang et al., 2016) and (3) Squeeze-and-Excitation block (Hu et al., 2017). The details of those architectures are included in the Appendix A. The results are presented in Table 4. Clearly, combined with the advantageous techniques/architectures, the performance of DSGC in image classificationcan can be further improved. It demonstrates that the DSGC can easily enjoy the benefit of the traditional 2d-convolution network development.

| Dataset | Subsampled Graphs | | | Original Graphs | | |
|---|---|---|---|---|---|---|
| | CIFAR10 | CIFAR100 | P | CIFAR10 | CIFAR100 | P |
| DSGC-VGG13 | 18.72% | 44.33% | 5.7M | 7.31% | 27.29% | 9.9M |
| DSGC-INCEPTION | 18.27% | 43.41% | 9.9M | **6.44**% | 28.55% | 12M |
| DSGC-DenseNet | 17.17% | 43.34% | 2.7M | 7.14% | **26.50**% | 2.9M |
| DSGC-SE | 18.71% | 44.15% | 6.1M | 7.00% | 27.26% | 10M |
| CNN | 18.03% | 43.42% | 18M | 6.86% | 26.86% | 18M |
| Xception | **17.07**% | **41.54**% | 3.1M | 7.08% | 26.84% | 3.1M |

Table 4: Summary of error rate on the test set in different settings.

## 4.6 TRAINING TIME COMPARISON

In table 5, we report the mean training time per epoch for DSGC and GCN, the fastest graph convolution baseline. In DSGC, our model computes the convolution weight for each edge of the graph, which requires more computation resources. However, we always perform the graph convolution on the sparse graph, which the number of edges grows only linearly in the graph size. Therefore the training is fairly efficient. Notably, learning the convolution filters as in DSGC leads to consistently better performance over all previous methods, with around 0.5x-3x running time.

| Dataset | CIFAR | USHCN-TMAX | 20news |
|---------|-------|------------|--------|
| GCN     | 1.75  | 0.465      | 0.207  |
| DSGC    | 3.81  | 1.73       | 0.280  |

Table 5: Training time per epoch for GCN and DSGC methods. The unit is minute.

## 5 RELATED WORK

In this section, we will summarize the graph convolution methods proposed in recent years with the label propagation process, which reveals the difference between traditional 2D-convolution and them. Firstly, we provide the formulation of the full convolution (LeCun et al., 1995),

$$y_{iq} = \sum_{p=1}^{P} \sum_{j \in \mathcal{N}(i)} w_{\Delta_{ij}}^{(pq)} x_{jp} \tag{6}$$

different from the depthwise separable convolution, it captures the channel correlation and spatial correlation simultaneously by $\boldsymbol{W}^{(pq)}$, which leads to the larger number of parameters.

In Spectral Network (Bruna et al., 2013), the authors try to leverage the graph Fourier transformation as the basic convolution operation in the graph domain, which can be written as,

$$y_{iq} = \sum_{p=1}^{P} \sum_{j \in \mathcal{N}(i)} w_{ij}^{(pq)} x_{jp} \quad where \quad \boldsymbol{W}^{pq} = \Phi \Lambda^{pq} \Phi^{T} \tag{7}$$

where $\Phi \in \mathbb{R}^{n \times n}$ contains the eigenvectors of Laplacian matrix of the graph, and $\Lambda$ is a diagonal matrix and learned by the supervision data. The Spectral Network can be matched with the full convolution, but with the different filter subspace, in other words, with different basic filters. However, it suffers from several limitations. (1) It needs to conduct eigenvector decomposition over the Laplacian Matrix, which is a very expensive operation. (2) The filters are not localized in the spatial domain. (3) The number of parameters grows linearly with the number of nodes in the graph. In order to address the previous problems, researchers try to use the Chebyshev polynomial to approximate the non-parameter filter $\Lambda$, which is referred to as ChebyNet (Defferrard et al., 2016). It can be written as,

$$y_{iq} = \sum_{k=1}^{K} \sum_{j \in \mathcal{N}(i)} T_k(L)_{ij} z_{iq}^{(k)} \quad where \quad \boldsymbol{z}_i^{(k)} = \boldsymbol{U}^{(k)} \boldsymbol{x}_j \tag{8}$$

where $T_k(L)$ is the $k$-th order Chebyshev polynomial term. The ChebyNet can be considered as the integration of $K$ depthwise separable convolution components in a layer. But still, it suffers from the similar limitation as the GCN, which is using one graph filter over all channels and the graph filter is constant given the input. So its model capacity still cannot compare with depthwise separable convolution. With larger $K$, the ChebyNet can approximate the non-parameter filers in the Spectral Network. However, it would require large number of parameters and face the similar limitation as the Spectral Network.

Besides the graph convolution methods, researchers propose another type of models, geometric convolution methods (Masci et al., 2015; Boscaini et al., 2016; Monti et al., 2016), to deal with data in the general spatial domain. Here, we introduce the most advanced one, MoNet (Monti et al., 2016) framework, which is also the most related one to our paper. The updating formula of MoNet in the label propagation process is,

$$y_{iq} = \sum_{k=1}^{K} \sum_{j \in \mathcal{N}(i)} w_k(v(i,j)) z_{jq}^{(k)} \quad where \quad \boldsymbol{z}_j^{(k)} = \boldsymbol{U}^k \boldsymbol{x}_j \tag{9}$$

where $w_k(v) = exp(-\frac{1}{2}(v - \mu_k)^T \Sigma_k^{-1}(v - \mu_k))$, and $v(i,j)$ is a mapping from a node pair to a embedding vector, similar to $\Delta_{ij}$ in our model. $\mu_k, \Sigma_k$ are both model parameters, and $\Sigma_k$ is constrained as the diagonal matrix. MoNet can be viewed as an extension of the ChebyNet by letting the graph filters learn from the data. But it still has two limitations compared with the depthwise separable convolution and proposed method: (1) It uses a simple Gaussian function, which is weaker than non-parametric filter in the depthwise separable convolution, and neural network function in the proposed method. (2) It uses a graph filter for all channels. In order to capture complex propagation patterns in a layer, the model requires a larger $K$, which leads to much larger number of parameters. And finally the experiment results show that the proposed method (DSGC) consistently outperforms the MoNet with less parameters in multiple tasks.

## 6 CONCLUSION

In this paper, we propose a novel Depthwise Separable Graph Convolution (DSGC) Network which is explicitly generalized from the depthwise separable convolution, and goes beyond to the general graph space. The extensive experiments on multi-field benchmark datasets demonstrate that our method can outperform strong baseline methods with a relatively small number of model parameters, and that it can be easily extended to leverage the advanced techniques/architectures in standard convolution networks for further improvement of the performance. In future work, we want to explore its impact on a broader range of applications, such as social networks and molecular structures by leveraging technical improvements about node/edge embedding based on graph structure information.

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

| Layers | VGG13 | DSGC-VGG13 | DSGC-DenseNet |
|---|---|---|---|
| Convolution | [ 3 × 3 conv ] × 2 | [ 9-conv ] × 2 | [ 9-conv ] × 6 |
| Transition | | | 1-conv |
| Pooling | 2 × 2 max-pooling | 4 max-pooling | |
| Convolution | [ 3 × 3 conv ] × 2 | [ 9-conv ] × 2 | [ 9-conv ] × 12 |
| Transition | | | 1-conv |
| Pooling | 2 × 2 max-pooling | 4 max-pooling | |
| Convolution | [ 3 × 3 conv ] × 2 | [ 9-conv ] × 2 | [ 9-conv ] × 24 |
| Transition | | | 1-conv |
| Pooling | 2 × 2 max-pooling | 4 max-pooling | |
| Convolution | [ 3 × 3 conv ] × 2 | [ 9-conv ] × 2 | [ 9-conv ] × 16 |
| Transition | | | 1-conv |
| Pooling | 2 × 2 max-pooling | 4 max-pooling | |
| Convolution | [ 3 × 3 conv ] × 2 | [ 9-conv ] × 2 | |
| Pooling | 2 × 2 max-pooling | 4 max-pooling | |
| Classifier | 512D fully-connected, softmax | | |

Table 6: Neural Network architecture for CIFAR datasets. Please see the text for more details.

# A  EXPERIMENT DETAIL

## A.1  IMPLEMENTATION DETAILS OF CIFAR EXPERIMENT

In section 4.2 and 4.5, we conduct the experiment on the CIFAR10 and CIFAR100 datasets. We will introduce the architecture settings for the DSGC and baseline models. Table 6 illustrates the basic architecture used in the experiment. In the DSGC-VGG13 and DSGC-DenseNet models, the $k$-conv refers to the spatial convolution (Eq.4) with $k$-nearest neighbors as the neighbor setting. So the 1-conv is the same as the $1 \times 1$ conv, which is doing linear transformation on channels. The hidden dimensions of VGG13 and DSGC-VGG13 are set as $\{256, 512, 512, 512\}$ and $\{256, 512, 512, 1024\}$. The growth rate of DSGC-DenseNet is 32. And the baseline graph and geometric convolution methods use the identical architecture as DSGC-VGG13. For the subsampled CIFAR experiment, We eliminate the first convolution, transition and pooling layer, and change the spatial convolution from 9-conv to $\{16\text{-conv}, 12\text{-conv}, 8\text{-conv}, 4\text{-conv}\}$. For the DSGC-SE, we follow the method described in Hu et al. (2017) to add the SE block to DSGC-VGG13 architecture. We use the dropout scheme described in Huang et al. (2016) for the DSGC-DenseNet model, and add the dropout layer after the pooling layer for VGG13 and DSGC-VGG13 models. For the DSGC-Inception model, we imitate the design of the Inception Network (Szegedy et al. (2016)). The key idea is letting a convolution layer have different size of convolution filters. We use a simple example as our Inception module, which is illustrated in Figure 3.

For the CNN model, we still format the input signal in the matrix shape. The signals in invalid points are set as 0. Furthermore, to perform the fair comparison with standard CNN in the subsampled situation, we append a mask matrix as an additional channel for input signals to indicate whether the pixel is valid or not. For the MoNet, we also apply the softmax trick described in Section 3, which accelerates its training process and improves its final result. For the ChebyNet, we set the polynomial order as $K = 3$.

For the $\triangle_{ij}$ used in DSGC and MoNet, we use a 5 dimension feature vector. We denote the coordinate of $i$-th node as $(x_i, y_i)$, and $\triangle x_{ij} = x_i - x_j, \triangle y_{ij} = y_i - y_j, \triangle d_{ij} = \triangle x_{ij}^2 + \triangle y_{ij}^2$. Then $\triangle_{ij} = (sign(\triangle x_{ij}), |\triangle x_{ij}|, sign(\triangle y_{ij}), |\triangle y_{ij}|, \triangle d_{ij})$.

The same learning schedule is applied to all models. We use SGD to train the model for 400 epochs. The initial learning rate is 0.1, and is divided by 10 at 50% and 75% of the total number of training epochs.

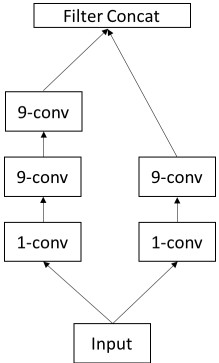

Figure 3: Inception Module

## A.2 IMPLEMENTATION DETAILS OF TIME SERIES PREDICTION

Firstly, we will give the formal definition of the time series forecasting, that is, spatiotemporal regression problem. We formulate the the spatiotemporal regression problem as a multivariate time series forecasting task with the sensors' location as the input. More formally, given a series of time series signals observed from sensors $\boldsymbol{Y} = \{\boldsymbol{y}_1, \boldsymbol{y}_2, \cdots, \boldsymbol{y}_T\}$ where $\boldsymbol{y}_t \in \mathbb{R}^n$ and $n$ are the number of sensors, and the locations of sensors $\boldsymbol{L} = \{\boldsymbol{l}_1, \boldsymbol{l}_2, \cdots, \boldsymbol{l}_n\}$ where $\boldsymbol{l}_i \in \mathbb{R}^2$ and indicates the coordinate of the sensor, the task is to predict a series of future signals in a rolling forecasting fashion. That being said, to predict $\boldsymbol{y}_{T+h}$ where $h$ is the desirable horizon ahead of the current time stamp $T$, we assume $\{\boldsymbol{y}_1, \boldsymbol{y}_2, \cdots, \boldsymbol{y}_T\}$ are available. Likewise, to predict the signal of the next time stamp $\boldsymbol{y}_{T+h+1}$, we assume $\{\boldsymbol{y}_1, \boldsymbol{y}_2, \cdots, \boldsymbol{y}_T, \boldsymbol{y}_{T+1}\}$ are available. In this paper, we follow the setting of the autoregressive model. Define a window size $p$ which is a hyper-parameter firstly. The model input at time stamp $T$ is $\boldsymbol{X}_T = \{\boldsymbol{y}_{T-p+1}, \cdots, \boldsymbol{y}_T\} \in \mathbb{R}^{n \times p}$. In the experiments of this paper, the horizon is always set as 1.

Intuitively, different sensors may have node-level hidden features to influence its propagation patterns and final outputs. Then for each node, the model learns a node embedding vector and concatenate it with the input signals. By using this trick, each node has limited freedom to interface with its propagation patterns. This trick is proven to be useful in this task, USHCN-PRCP and Solar specifically. We set the embedding size as 10 for these two datasets.

One thing readers may notice is that there are 10% data in USHCN dataset missing. To deal with that, we add an additional feature channel to indicate which point is missing. For the time series models, we tune the historical window $p$ according to the validation set. For the rest of models, we set the window size $p = 18$ for Solar dataset and $p = 6$ for USHCN datasets. The network architecture used in this task is 7 convolution layers followed by a regression layer. The $\triangle_{ij}$ setting is the same as the previous one. We use the Adam optimizer (Kingma & Ba, 2014) for this task, and train each model 200 epochs with learning rate 0.001.

## A.3 IMPLEMENTATION DETAILS OF DOCUMENT CATEGORIZATION

The data preprocessing follows the experiment details in Defferrard et al. (2016). And the network architecture for all models is 5 convolution layers followed by two MLP layers as the classifier. After each convolution layer, a dropout layer is performed with dropout rate of 0.5. The nodes' coordinate is the word embedding, and the method to calculate $\triangle_{ij}$ is similar to the previous ones. The optimizer used in this task is the same as the CIFAR experiment.

## A.4 VARIANCE OF DSGC PERFORMANCE

In this section, we report the variance of DSGC method in all 3 tasks. We run the DSGC model for 10 times and report the mean±std: CIFAR $7.39 \pm 0.136$, USHCN-TMAX $5.211 \pm 0.0498$, 20news $71.70 \pm 0.285$. Obviously, the variance is significantly smaller than the performance gap between the DSGC model and best baseline results (CIFAR 8.34, USHCN-TMAX 5.467, 20news 71.01).

