# OpenReview forum: "Learning Graph Convolution Filters from Data Manifold"
_ICLR.cc/2018/Conference — Reject_

### Official Review · AnonReviewer2 · 2017-11-27
**Depthwise convolution for GCN, seems to improve performance but requires more work**

**Rating:** 4
**Confidence:** 5

**Review:**

The paper presents a Depthwise Separable Graph Convolution network that aims
at generalizing Depthwise convolutions, that exhibit a nice performance in image
related tasks, to the graph domain. In particular it targets
Graph Convolutional Networks.

In the abstract the authors mention that the Depthwise Separable Graph Convolution
that they propose is the key to understand the connections between geometric
convolution methods and traditional 2D ones. I am afraid I have to disagree as
the proposed approach is not giving any better understanding of what needs to be
done and why. It is an efficient way to mimic what has worked so far for the planar
domain but I would not consider it as fundamental in "closing the gap".

I feel that the text is often redundant and that it could be simplified a lot.
For example the authors state in various parts that DSC does not work on
non-Euclidean data. Section 2 should be clearer and used to better explain
related approaches to motivate the proposed one.
In fact, the entire motivation, at least for me, never went beyond the simple fact
that this happens to be a good way to improve performance. The intuition given
is not sufficient to substantiate some of the claims on generality and understanding
of graph based DL.

In 3.1, at point (2), the authors mention that DSC filters are learned from the
data whereas GC uses a constant matrix. This is not correct, as also reported in
equation 2. The matrix U is learned from the data as well.

Equation (4) shows that the proposed approach would weight Q different GC
layers. In practical terms this is a linear combination of these graph
convolutional layers.
What is not clear is the \Delta_{ij} definition. It is first introduced in 2.3
and described as the relative position of pixel i and pixel j on the image, but
then used in the context of a graph in (4). What is the coordinate system used
by the authors in this case? This is a very important point that should be made
clearer.

Why is the Related Work section at the end? I would put it at the front.

The experiments compare with the recent relevant literature. I think that having
less number of parameters is a good thing in this setting as the data is scarce,
however I would like to see a more in-depth comparison with respect to the number
of features produced by the model itself. For example GCN has a representation
space (latent) much smaller than DSCG.
No statistics over multiple runs are reported, and given the high variance of
results on these datasets I would like them to be reported.

I think the separability of the filters in this case brings the right level of
simplification to the learning task, however as it also holds for the planar case
it is not clear whether this is necessarily the best way forward.
What are the underlying mathematical insights that lead towards selecting
separable convolutions?

Overall I found the paper interesting but not ground-breaking. A nice application
of the separable principle to GCN. Results are also interesting but should be
further verified by multiple runs.

---

> ### Author Response · Authors · 2017-12-11
> **Rebuttal**
>
> Thank you for the comments. Following are our responses to the major points:
>
>
> “the Depthwise Separable Graph Convolution that they propose is the key to understand the connections between geometric convolution methods and traditional 2D ones.”
> Let us clarify: the key insight in our paper is that many successful 2d-grid based CNNs (including DSC) cannot directly apply to generic spatial convolution problems, and we close the gap (in a mathematically compatible way) by proposing a unified framework for both 2d-grid based convolution and for more generic spatial convolution with automatically learning link weights for the underlying graphs. It is not our claim that DSGC is the only way to close the gap. We apologize if our wording in the Abstract is not clear enough; we will make it clear in the revised version.
>
>
> “the authors mention that DSC filters are learned from the data whereas GC uses a constant matrix. This is not correct,”
> Apologies for the confusion. Following the terminology in [1], DSC consists of two parts, i.e., the spatial convolution (W in eq.3) and the channel convolution (U in eq.3). What we really mean here is that  GC learns the channel convolution but relies on a constant filter to perform the spatial convolution. On the other hand, DSC learns both the spatial convolution and the channel convolution. Further, DSGC generalizes the DSC method.
>
> “In practical terms this is a linear combination of these graph convolutional layers.”
> Just to clarify, we are not learning to combine GC layers, but learning the filter weights associated with the edges in each graph. The resulting operation is not a simple linear combination of GC layers. This can be read from eq. (4), where the summation is carried out over edges/neighbors instead of over layers. It learns graph spatial filters, while conventional GC is not able to do.
>
>
> “What are the underlying mathematical insights that lead towards selecting separable convolutions?”
> “It is an efficient way to mimic what has worked so far for the planar domain but I would not consider it as fundamental in "closing the gap"
> How to generalize standard convolution over a 2d grid to the general spatial domain is the fundamental problem we are trying to address and the major contribution of our paper. Existing techniques such as traditional graph convolution (GC) is not compatible with grid-based convolution as it uses the constant spatial filter across all channels. Our approach provides a natural mechanism to close the gap by learning a separable convolution filter for different channels using function approximation. Besides, the effectiveness of our approach was empirically verified by our experiments using datasets over a variety of domains.
>
> “What is not clear is the \Delta_{ij} definition. It is first introduced in 2.3 and described as the relative position of pixel i and pixel j on the image, but then used in the context of a graph in (4)..”
> Given a pair of nodes i and j, Delta_{ij} can be viewed as the embedding of its spatial attributes (e.g. the relative difference between the two nodes’ spatial coordinates), which is needed for MLPs to predict the filter weights. In other words, \Delta_{ij} serves as a “key” to retrieve “values” (filter weights) either from a lookup table as in 2d-grid convolution (sec 2.4) or from a compressed table as in DSGC (MLP in (4)). As stated in the introduction, we did not explore deeper in graph systems without spatial coordinate information, although our model can subsume GC with the certain manually defined coordinate system (see [2] for more detailed discussions). We will make these points more explicit in the revised paper.
>
> “Results are also interesting but should be further verified by multiple runs.”
> We agree reporting variance would further strengthen the paper, and will add such results in our revision. Actually, we did not observe high variance performance in our experiment. We rerun the DSGC model for 10 times and report the mean(std error) in three tasks: CIFAR 7.39(0.136), USHCN-TMAX 5.211(0.0498), 20news 71.70(0.285). Obviously, the variance is significantly smaller than the performance gap between the DSGC model and best baseline results (CIFAR 8.34, USHCN-TMAX 5.467, 20news 71.01).
>
>
> [1] Chollet, François. "Xception: Deep Learning with Depthwise Separable Convolutions." arXiv preprint arXiv:1610.02357(2016).
>
> [2] Monti, Federico, Davide Boscaini, Jonathan Masci, Emanuele Rodolà, Jan Svoboda, and Michael M. Bronstein. "Geometric deep learning on graphs and manifolds using mixture model CNNs." arXiv preprint arXiv:1611.08402 (2016).

---

### Official Review · AnonReviewer1 · 2017-11-27
**Extension of Depth-Wise-Convolution (Chollet et al. 2016) with improved performance.**

**Rating:** 6
**Confidence:** 4

**Review:**

The paper presents an extension of the Xception network of (Chollet et al. 2016) 2D grids to generic graphs. The Xception network decouples the spatial correlations from depth channels correlations by having separate weights for each depth channel. The weights within a depth channel is shared thus maintaining the stationary requirement. The proposed filter relaxes this requirement by forming the weights as the output of a two-layer perception.

The paper includes a detailed comparison of the existing formulations from the traditional label propagation scheme to more recent more graph convolutions (Kipf & Welling, 2016 ) and geometric convolutions  (Monti et al. 2016).

The paper provides quantitative evaluations under three different settings i) image classification, ii) Time series forcasting iii) Document classification. The proposed method out-performs all other graph convolutions on all the tasks (except image classification) though having comparable or less number of parameters. For image classification, the performance of proposed method is below its predecessor Xception network.

Pros:
i) Detailed review of the existing work and comparison with the proposed work.
ii) The three experiments performed showed variety in terms of underlying graph structure hence provides a thorough evaluation of different methods under different settings.
iii) Superior performance with fewer number of parameters compared to other methods.
Cons:
i) The architecture of the 2 layer MLP used to learn weights for a particular depth channel is not provided.
ii) The performance difference between Xception and proposed method for image classification experiments using CIFAR is incoherent with the intuitions provided Sec 3.1 as the proposed method have more parameters and is a generalized version of DSC.

---

> ### Author Response · Authors · 2017-12-11
> **Rebuttal**
>
> Thank you for the comments. Following are our responses for the major points:
>
> “The architecture of the 2 layer MLP used to learn weights for a particular depth channel is not provided.”
>
> We will add the details in the revised version: The 2 layer MLP takes the \Delta_{ij} as the input. The hidden dimension is 256 with tanh activation. The output dimension is 1. Parameters of the MLPs are learned independently for each filter. We have conducted an ablation study for the MLP, by changing their depth, activation functions, and weight sharing strategies. However, their results are very similar; the two-layer MLP provides the reasonable performance with the shortest running time.
>
> “The performance difference between Xception and proposed method for image classification experiments using CIFAR is incoherent with the intuitions provided Sec 3.1 as the proposed method have more parameters and is a generalized version of DSC.”
>
> By “incoherent” do you mean that DSGC (our proposed method) should always outperform the original DSC in image classification? This may not be necessarily true and is not our expectation. That is, when the underlying structure is a truly 2d grid (like images), the simpler DSC model would fit the problem better and hence is expected to outperform the more generalized model of DSGC. On the other hand, when the true underlying structure is not a 2d grid (as in many graph convolution problems), then the DSGC is more powerful, as we have shown in our experimental results for the spatiotemporal modeling tasks.

---

### Official Review · AnonReviewer3 · 2017-11-28
**Incremental yet interesting advance in geometric CNNs. But, some core technical aspects and experiments are missing.**

**Rating:** 5
**Confidence:** 3

**Review:**

Paper Summary:
This work proposes a new geometric CNN model to process spatially sparse data. Like several existing geometric CNNs, convolutions are performed on each point using nearest neighbors. Instead of using a fixed or Gaussian parametric filters, this work proposes to predict filter weights using a multi-layer perception. Experiments on 3 different tasks showcase the potential of the proposed method.

Paper Strengths:
- An incremental yet interesting advance in geometric CNNs.
- Experiments on three different tasks indicating the potential of the proposed technique.

Major Weaknesses:
- Some important technical details about the proposed technique and networks is missing in the paper. It is not clear whether a different MLP is used for different channels and for different layers, to predict the filter weights. Also, it is not clear how the graph nodes and connectivity changes after the max-pooling operation.
- Since filter weight prediction forms the central contribution of this work, I would expect some ablation studies on the MLP (network architecture, placement, weight sharing etc.) that predicts filter weights. But, this is clearly missing in the paper.
- If one needs to run an MLP for each edge in a graph, for each channel and for each layer, the computation complexity seems quite high for the proposed network. Also, finding nearest neighbors takes time on large graphs. How does the proposed technique compare to existing methods in terms of runtime?

Minor Weaknesses:
- Since this paper is closely related to Monti et al., it would be good if authors used one or two same benchmarks as in Monti et al. for the comparisons. Why authors choose different set of benchmarks? Because of different benchmarks, it is not clear whether the performance improvements are due to technical improvements or sub-optimal parameters/training for the baseline methods.
- I am not an expert in this area. But, the chosen benchmarks and datasets seem to be not very standard for evaluating geometric CNNs.
- The technical novelty seems incremental (but interesting) with respect to existing methods.

Clarifications:
- See the above mentioned clarification issues in 'major weaknesses'. Those clarification issues are important to address.
- 'Non-parametric filter' may not be right word as this work also uses a parametric neural network to estimate filter weights?

Suggestions:
- It would be great if authors can add more details of the multi-layer perceptron, used for predicting weights, in the paper. It seems some of the details are in Appendix-A. It would be better if authors move the important details of the technique and also some important experimental details to the main paper.

Review Summary:
The proposed technique is interesting and the experiments indicate its superior performance over existing techniques. Some incomplete technical details and non-standard benchmarks makes this not completely ready for publication.

---

> ### Author Response · Authors · 2017-12-11
> **Rebuttal Part(1/2)**
>
> We thank the reviewer for the comments/questions. Following are our main clarifications:
>
> “Some important technical details about the proposed technique and networks is missing in the paper. It is not clear whether a different MLP is used for different channels and for different layers, to predict the filter weights. Also, it is not clear how the graph nodes and connectivity changes after the max-pooling operation.”
>
> We consider the most generic setup where each filter comes with its own MLP (eq. 4, w^{(q)} refers different function.), although partially sharing those MLPs could be a potential option. The pooling layer is performed based on k-means clustering, namely, nodes in the previous layer before pooling will be connected to their cluster centroid in the next layer after pooling (described in Section 4.1). After pooling, edges in the graph are still defined based on k-nearest neighbors. We will make these points more clearly in the revisited version.
>
> “Since filter weight prediction forms the central contribution of this work, I would expect some ablation studies on the MLP (network architecture, placement, weight sharing etc.) that predicts filter weights. “
>
> We have indeed conducted ablation tests with MLP, by changing the number of layers and activation function of each hidden layer, and by trying several weight sharing strategies. The results are very similar in terms of accuracy; the two-layer MLP provides a reasonable performance with the shortest running time and hence is used in the current paper. We chose not to report more results on our preliminary ablation study because we don’t have enough mathematical understanding about the influence of different MLP architectures to the final performance, which makes the design of ablation experiments very subjective. We will include those details in the appendix of the revisited paper.
>
> “If one needs to run an MLP for each edge in a graph, for each channel and for each layer, the computation complexity seems quite high for the proposed network. Also, finding nearest neighbors takes time on large graphs. How does the proposed technique compare to existing methods in terms of runtime?”
>
> The number of edges grows only linearly in the graph size, i.e., the number of nodes, because of the sparsity of the graph. Therefore the training is fairly efficient. Also note that the nearest neighbor computation can be carried out during the preprocessing, hence does not affect the training time. We will provide the detailed information in our revised version of the paper for comparing the running time  of all graph convolution algorithms, as shown below: m is for minutes
> Cifar
> Dcnn 922m	Chebynet 1715.71m	GCN 706m	MoNet  2504m	DSGC 1527m
> TIme series prediction
> Dcnn 176m	Chebynet 286m 	GCN 93m 	MoNet  620m 	DSGC 346m
> Document Classification
> Dcnn 158m 	Chebynet 278m 	GCN 83m 	MoNet  842m 	DSGC 112m
> Notably, learning the convolution filters as in DSGC leads to consistently better performance over all previous methods, with around 0.5x-3x running time.

---

> > ### Author Response · Authors · 2017-12-11
> > **Rebuttal part(2/2)**
> >
> > “the chosen benchmarks and datasets seem to be not very standard for evaluating geometric CNNs.”
> >
> > We use benchmark datasets for image classification (CIFAR) and document categorization (20news), following ([1],[2],[3]). As you mentioned, Monti et al. [3] used MNIST and citation network as the experiment datasets. We use CIFAR instead of MNIST as the former is more difficult and can better demonstrate the scalability of our algorithm. Moreover, we chose not to use some “standard” citation networks as their training sets are extremely small, e.g. 140 samples for Cora and 60 samples for Pubmed in their experiment setting, which usually lead to unreliable results, as pointed out by Monti et al. \cite{3}, “The tuning of the network hyper-parameters has been fundamental in this case for avoiding overfitting, due to a very small size of the training set.” Finally, the spatio-temporal forecasting is a valuable application for the graph convolution method ([4],[5]).
> >
> >
> > “The technical novelty seems incremental (but interesting) with respect to existing methods.”
> >
> > One major novel contribution is to provide a unified mathematical view for both 2d-grid based convolution methods and more generic graph convolution, which has not been done before. Since our approach is fully compatible with 2d-grid convolution, it would enable people to better leverage architectures and techniques developed for 2d-grid convolution with mathematical understanding, not just intuition.
> >
> > “‘'Non-parametric filter' may not be right word as this work also uses a parametric neural network to estimate filter weights?”
> >
> > Do you refer this sentence “which is weaker than non-parametric filter in the depthwise separable convolution and neural network function in the proposed method”? Hereby “non-parametric filter” we mean the standard (2d-grid based) depthwise separable convolution, and by “neural network function” we mean the same as in your words. Sorry for the confusion in wording; we will rephrase this sentence to make it clear.
> >
> > “It would be great if authors can add more details of the multi-layer perceptron, used for predicting weights, in the paper. It seems some of the details are in Appendix-A. It would be better if authors move the important details of the technique and also some important experimental details to the main paper.”
> >
> > We will do that.  Thanks for the suggestion.
> >
> > [1] Bruna, Joan, Wojciech Zaremba, Arthur Szlam, and Yann Lecun. "Spectral networks and locally connected networks on graphs." In International Conference on Learning Representations (ICLR2014), CBLS, April 2014. 2014.
> > [2] Defferrard, Michaël, Xavier Bresson, and Pierre Vandergheynst. "Convolutional neural networks on graphs with fast localized spectral filtering." In Advances in Neural Information Processing Systems, pp. 3844-3852. 2016.
> > [3] Monti, Federico, Davide Boscaini, Jonathan Masci, Emanuele Rodolà, Jan Svoboda, and Michael M. Bronstein. "Geometric deep learning on graphs and manifolds using mixture model CNNs." arXiv preprint arXiv:1611.08402 (2016).
> > [4] Li, Yaguang, Rose Yu, Cyrus Shahabi, and Yan Liu. "Graph Convolutional Recurrent Neural Network: Data-Driven Traffic Forecasting." arXiv preprint arXiv:1707.01926 (2017).
> > [5] Yu, Bing, Haoteng Yin, and Zhanxing Zhu. "Spatio-temporal Graph Convolutional Neural Network: A Deep Learning Framework for Traffic Forecasting." arXiv preprint arXiv:1709.04875 (2017).

---

### Decision · Program_Chairs · 2018-01-29
**ICLR 2018 Conference Acceptance Decision**

**Decision:**

Reject

**Comment:**

This paper proposes to combine Depthwise separable convolutions developed for 2d grids with recent graph convolutional architectures. The resulting architecture can be seen as learning both node and edge features, the latter encoding node similarities with learnt weights.
Reviewers agreed that this is an interesting line of work, but that further work is needed in both the presentation and the experimental front before publication. In particular, the paper should also compare against recent models (such as the MPNN from Gilmer et al) that also propose edge feature learning. THerefore, the AC recommends rejection at this time.